# Enhancing TB Vaccine Efficacy: Current Progress on Vaccines, Adjuvants and Immunization Strategies

**DOI:** 10.3390/vaccines12010038

**Published:** 2023-12-29

**Authors:** Hui Wang, Shuxian Wang, Ren Fang, Xiaotian Li, Jiayin Xing, Zhaoli Li, Ningning Song

**Affiliations:** 1Weifang Key Laboratory of Respiratory Tract Pathogens and Drug Therapy, School of Life Science and Technology, Shandong Second Medical University, Weifang 261053, China; 15643967892@163.com (H.W.); wangshuxian9812@163.com (S.W.); 18845750439@163.com (R.F.); 18370358360@163.com (X.L.); 15703547386@163.com (J.X.); 2SAFE Pharmaceutical Technology Co., Ltd., Beijing 100000, China

**Keywords:** TB, vaccine, adjuvant

## Abstract

Tuberculosis (TB) remains a global infectious disease primarily transmitted via respiratory tract infection. Presently, vaccination stands as the primary method for TB prevention, predominantly reliant on the Bacillus Calmette–Guérin (BCG) vaccine. Although it is effective in preventing disseminated diseases in children, its impact on adults is limited. To broaden vaccine protection, efforts are underway to accelerate the development of new TB vaccines. However, challenges arise due to the limited immunogenicity and safety of these vaccines, necessitating adjuvants to bolster their ability to elicit a robust immune response for improved and safer immunization. These adjuvants function by augmenting cellular and humoral immunity against *M. tuberculosis* antigens via different delivery systems, ultimately enhancing vaccine efficacy. Therefore, this paper reviews and summarizes the current research progress on *M. tuberculosis* vaccines and their associated adjuvants, aiming to provide a valuable reference for the development of novel TB vaccines and the screening of adjuvants.

## 1. Introduction

Tuberculosis (TB) is a significant global infectious disease responsible for numerous fatalities worldwide. *Mycobacterium tuberculosis* (Mtb) infects through the respiratory tract and is phagocytosed by alveolar macrophages, where it multiplies. An anti-TB immune response leads to the formation of TB granulomas. Although the immune response inhibits Mtb proliferation, not all Mtb are eradicated; some persist and can reactivate when the immune system weakens. According to the latest WHO report, there were ~10.6 million TB patients and ~1.3 million TB-related deaths in 2022. The prolonged TB treatment regimen has led to increasing drug resistance due to the extensive use of antibiotics [1]. In 2022, there were 410,000 cases of multidrug-resistant/rifampicin-resistant TB worldwide. The continuous emergence of multi-drug-resistant and extensively drug-resistant strains poses significant challenges to TB prevention and treatment, necessitating urgent research and development of new anti-tuberculosis vaccines and drugs.

Vaccination is a crucial method for TB prevention, as it stimulates the immune system to produce antibodies against the bacterium, thus safeguarding the body from TB. BCG provides moderately good protection for infants and young children against severe forms of TB but lacks effective protection against the main transmission population of TB infection and disease burden—adolescents and adults. Developing a new and suitable vaccine for all ages that is effective for all forms of TB is the key element to curbing TB. The generation of new TB cases can be effectively avoided by vaccinating adolescents and adults [2]. The findings suggest that a TB vaccine can significantly reduce TB morbidity and mortality, and the vaccine for adolescents and adults has a greater impact than a vaccine for infants [3]. Therefore, vaccine safety is paramount, and currently, safer subunit vaccines, which only contain a portion of the Mtb antigen and require an appropriate adjuvant for enhanced immune response, are preferred.

## 2. Research Progress of the TB Vaccine

The BCG vaccine, although effective in safeguarding infants and young children against TB, lacks efficacy in adolescents and adults and does not treat infected individuals. Moreover, due to the rise in multi-drug-resistant Mtb strains, its protective impact diminishes over time, rendering it essentially ineffective for adults [4]. Given BCG’s limitations, there is an urgent necessity for a more comprehensive and long-lasting vaccine. The latest WHO Global Tuberculosis Report identifies 16 vaccine candidates in clinical stages, categorized by their composition: viral vector, live attenuated, mRNA and subunit vaccines (Table 1).

### 2.1. Viral-Vector-Vaccine

Viral vector vaccines utilize diverse viruses as carriers to express protective antigens at high levels, stimulating cellular immune responses based on their immunogenicity. Currently, the main viral vectors in new TB vaccine development include adenoviral vectors, modified vaccinia virus Ankara (MVA) and vectored influenza virus vectors [5].

AdHu5Ag85A is a TB vaccine based on human adenovirus type 5 (AdHu5), which expresses the Ag85A antigen. Adenoviruses, which encompass both mammalian and avian genera, are double-stranded DNA viruses. Human adenoviruses comprise over 100 types, 6 subgroups (A–F) and over 50 serotypes. These viruses possess a stable genome that does not readily integrate into human genes, and they offer a range of serotypes suitable for gene delivery vectors in vaccine development [6]. Additionally, adenovirus serves as a highly promising vector for TB vaccines due to its natural ability to have high Ad tropism toward respiratory epithelial cells, strong immunogenicity, adjuvant activity and the capacity to stimulate CD8+ T cell immunity. Currently, the vaccine is in phase I clinical trials, demonstrating safety and tolerance when administered intramuscularly. It has also been proven to be more effective in augmenting CD4+ and CD8+ T cell immunity in individuals previously vaccinated with the BCG vaccine [7]. Furthermore, vaccination elicits Ag85A-specific T cell responses in the bloodstream, and low-dose aerosol vaccination triggers T cell responses and sustained transcriptional changes in alveolar macrophages. This suggests that aerosol administration is a safe and efficient means of inducing respiratory mucosal immunity, providing strong support for the development of aerosolized TB vaccines [8].

MVA85A, a live carrier vaccine combining MVA with Mtb major secreted protein antigen 85A, exhibits a favorable safety profile. Administering MVA85A after BCG vaccination proves more effective than MVA85A alone, thereby suggesting an enhanced immune response to pre-existing MTB antibodies [9]. This vaccine elicits and induces an immune response in antigen Ag85A T cells, resulting in robust and enduring CD4+ T cell proliferation and differentiation and increased IFN-γ, TNF-α, IL-2, IL-17 and GM-CSF secretions. Phase I trials evaluated its safety and immunogenicity, and the results showed that aerosol delivery can enhance MVA85A’s immunogenicity [10]. Moreover, the vaccine offered improved safety protection in adults with latent TB infection and demonstrated good tolerance and immunogenicity in healthy UK adults [11]. In phase II clinical trials, the vaccine was deemed safe for the primary immunization of HIV-exposed neonates, inducing an early and modest specific immune response without affecting subsequent BCG vaccination’s immunogenicity [12]. It remained highly immunoprotective, tolerable and immunogenic even in HIV-infected adult patients [13]. TB/FLU-05E attenuated influenza virus vector vaccines developed in Russia for prophylaxis have shown in animal experiments the ability to induce long-term cell-mediated immunity (CMI) and reduce the risk of Koch’s phenomenon in mice with latent TB infection (LTBI). Phase Ia trials are ongoing to assess safety, reactogenicity and immunogenicity [14].

### 2.2. Live Attenuated Vaccine

Live attenuated vaccines result from a series of treatments that significantly reduce pathogen virulence while preserving immunogenicity. This category includes BCG, VPM1002 and MTBVAC, with the latter two in phase III clinical trials. VPM1002 exhibits comparable immunogenicity to BCG, making it a promising BCG replacement.

VPM1002 is used for neonatal and post-exposure immunization against recurrent TB in adults. This live attenuated vaccine is primarily a genetic modification of BCG, enhancing immunity by replacing the *urease C* gene with the *Listeria monocytogenes-encoded listeriolysin O* (LLO) gene. Urease reduction in phagolysosomal acidification promotes Mtb survival [15]. Phase I clinical trials in Germany and South Africa confirmed its safety and immunogenicity in young individuals. Phase IIa studies in healthy neonates and phase IIb studies in HIV-exposed and unexposed neonates in South Africa demonstrated VPM1002′s good immunogenicity and safety [16]. A Phase II/III clinical trial in India aims to prevent relapse after standard TB treatment in adults. A phase III trial led by Blossey AM demonstrated that VPM1002 vaccination was well-tolerated by the elderly, with a preventive effect against severe respiratory disease in older adults [17].

MTBVAC, an attenuated live strain of Mtb, expresses most Mtb antigens [18]. It has two deleted virulence genes (*phoP* and *fadD26*) in Mtb but retains several T cell epitopes, including ESAT-6 (a major immunodominant BCG and CFP10 antigen). The deletion of *phoP* in MTBVAC results in ESAT-6 expression but not secretion [19]. MTBVAC stimulates type I conventional dendritic cells, enhancing tumor antigen-specific CD4 and CD8 T cell responses [20]. Currently, it is in phase III clinical trials in South Africa [21]. Spertin F et al. demonstrated that MTBVAC has a safety profile similar to BCG and offers more effective immune protection [22].

### 2.3. mRNA Vaccine

The mRNA vaccine stimulates an immune response by introducing mRNA to deliver specific antigens. These antigens encode mRNA molecules inducing both humoral and cellular immunity, enabling immune cells to produce proteins from the virus. These proteins stimulate an immune response that forms a protective barrier against infection, aiding in pathogen recognition and destruction [23]. Since mRNA vaccines are not constructed by live pathogens, they are non-infectious and offer multiple advantages, such as high effectiveness, easy adaptability, an easy antigen design and the capacity to induce both humoral and cellular immunity. Their promise in infectious disease vaccination is significant [24]. The rapid development of RNA-based vaccines has protected the global population from viruses like SARS-CoV-2, reducing economic losses. mRNA vaccines provide superior protection and stronger immune signals compared to traditional vaccines, offering prolonged humoral and cell-mediated immunity against the targeted antigen [25]. The success of mRNA vaccines in viral immunization has accelerated bacterial vaccine development, such as for Mtb [26,27]. Arsen et al. formulated an RNA vaccine against Mtb, utilizing the highly expressed CD4+ and CD8+ T cell epitope antigens. It aims to combat intracellular Mtb, enhance cellular and humoral immune responses and reduce bacterial loads in the lungs of mice via aerosol delivery. The TB mRNA vaccine is currently in clinical trials, and the BNT164a1/BNT164b1 vaccine from Germany is undergoing phase Ia trials in healthy adults aged 18–55 years [1].

### 2.4. Subunit Vaccine

Subunit vaccines comprise nucleic-acid-free, immunologically active antigens with a few specific epitopes that are crucial for protective immunity. By isolating and purifying proteins, these protective immune-response-associated specific antigens can be used to make subunit vaccines.

ID93 + GLA-SE includes four antigens: Rv1813, Rv2608, Rv3619 and Rv3620, alongside the GLA-SE liposomal adjuvant [28]. Phase IIa clinical trials on ID93 + GLA-SE revealed its ideal safety profile and high tolerability, proving that it is well tolerated and immunogenic in tuberculosis-treated adults. ID93 + GLA-SE triggers a multifunctional CD4+ T cell response, primarily generating IgG1 and IgG3 subclass antibodies [29]. Rv1813 is closely linked to latent Mtb infection, activating CD4+ T lymphocytes in latently infected individuals to release significant IFN-γ, thus potentially playing a protective role in their immune response. Rv2608, a member of the PPE family protein, is associated with virulence [30]. Rv3619 and Rv3620 are specific to Mtb, acting as secreted virulence-related proteins. Furthermore, these antigens are compatible with the GLA-SE adjuvant, where GLA serves as a toll-like receptor 4 (TLR4) agonist, and the oil-in-water emulsion SE functions as a delivery vehicle for GLA.

AEC/BC02, a recombinant tuberculosis vaccine containing the Mtb antigen Ag85b, ESAT6-CFP10 (EC) and the adjuvant system BC02, exhibited robust protection in a guinea pig model of latent infection. It reduced organ damage and decreased bacilli loads in the spleen and lungs [31]. Additionally, Ag85b immunization conferred protection against TB infection by increasing IFN-γ production. ESAT6 triggered a potent cellular immune response, and CFP10 activated in vivo CD8+ T cells with sustained cytolytic activity post-Mtb infection. The adjuvant BC02 is a mixture of BCG unmethylated cytosine-phosphate-guanine (CpG) DNA fragments and aluminum salt [32].

M72/AS01E is a subunit vaccine containing two Mtb antigens (MTB32A and MTB39A) with AS01E as an adjuvant [33]. MTB39A (Rv1196) from the PPE family inhibits NF-κB/rel-mediated pro-inflammatory cytokine production by up-regulating and phosphorylating the expression of suppressor of cellulose 3 (SOCS3) [34]. MTB32A (Rv0125) functions as a serine protease [35]. The AS01E adjuvant system comprises MPL, QS-21 and liposomes. In clinical trials, this vaccine was well tolerated by HIV-negative adults with latent TB infection, offering 54.0% protection against active TB disease in Mtb-infected adults [36]. Numerous studies and analyses have confirmed the safety and efficacy of the M72/AS01E vaccine. Despite mild adverse effects observed in follow-up visits, its potential in preventing TB in adults remains promising [37].

H56:IC31, comprising the IC31 adjuvant and three antigens (Ag85B, ESAT-6 and the latency-associated Rv2660c), enhances cellular and humoral immune responses [38]. The protective effect of the H56:IC31 vaccine is linked to the persistence and function of its induced CD4+ T cell response. Ag85B (an acetyl/branched acyltransferase involved in Mtb cell wall synthesis) and ESAT-6 exhibited high immunogenicity in both human and TB animal models by maintaining sustained high expression during Mtb infection [39].

The GamTBVac contains Ag85A, ESAT6-CFP10 and the CpG ODN adjuvant. Phase II clinical data demonstrate that this vaccine is well tolerated, inducing specific and durable Th1 and humoral immune responses [40]. The ESAT6, CFP10 and Ag85A antigens are associated with Mtb proliferation, demonstrating high immunogenicity and immunodominance. The CpG ODN adjuvant, a nucleic-acid-based adjuvant, effectively triggers a Th1 response, annihilating Mtb by stimulating toll-like receptors in mammalian immune cells, activating T lymphocytes and inducing IFN-γ secretion [41].

## 3. Adjuvant

Vaccines prepared from attenuated live pathogens are highly immunogenic and cost-effective to produce. However, large-scale applications often fail to meet demand due to long incubation periods and safety concerns. Currently, newer vaccines with improved safety profiles are increasingly replacing some attenuated live vaccines in clinical practice. But these modern vaccines elicit weaker immune responses when administered alone. Due to these limitations, adjuvants can help vaccines induce strong immune responses when co-delivered with antigens as non-specific immune enhancers. Adjuvants are capable of inducing and enhancing both cellular and humoral immune responses to *Mycobacterium tuberculosis* antigens through various delivery systems, thereby providing long-term protection. Vaccine adjuvants can be broadly classified into categories such as aluminum salts, proteins, nucleic acids, liposomes and others (Table 2). As adjuvants are key components of subunit vaccines, developing safe and effective novel adjuvants, as well as expanding the use of adjuvants, is crucial.

Aluminum salts, including aluminum hydroxide, aluminum phosphate and amorphous aluminum phosphate, are commonly used adjuvants with diverse applications. Aluminum potassium sulfate enhances the efficacy of the tuberculosis vaccine and prolongs the body’s immune response duration [42]. Aluminum hydroxide, as an adjuvant, activates IL-4 secretion by Th2 cells and induces the expression of CD83, CD86 and MHC-II molecules. This results in a Th2 humoral immune response and strengthens the immune effect of the TB vaccine [39]. Aluminum salts trigger NLRP3 inflammatory vesicle activation, promoting early IL-1β cytokine activation, which generates cellular immunity at the injection site. IL-1β also induces local inflammation, recruits antigen-presenting cells, facilitates dendritic cell maturation, enhances antigen uptake and stimulates T cells. Aluminum hydroxide acts as an adjuvant by activating the complement system, which can induce chronic inflammation by enhancing immune responses through B cells and dendritic cells [43]. BC02 comprises BCG-derived unmethylated CpG DNA fragments and Al(OH)3. CpG induces a Th1 immune response, and aluminum salts promote a Th2 response by secreting IL-4 and IL-5 cytokines and producing IgG1 and IgE antibodies. This enhances the immunological effect of the Mtb vaccine [31].

### 3.1. Protein Adjuvants

Protein adjuvants are biologically active substances secreted by stimulated cells. They are typically small peptides or glycoproteins that promote Th cell differentiation and enhance the function of NK cells and T lymphocytes while broadly regulating the immune response. When combined with Mtb antigens, the IC31 adjuvant induces a long-term Th1 immune response. IC31 is formed by complexing the antimicrobial peptides KLK and ODN1a. KLK is a cationic peptide, and ODN1a is a negatively charged DNA oligonucleotide that binds to TLR9, triggering antigen-specific Th1 or Th17 immune responses [44]. KLK facilitates the phagocytosis of ODN1a and its delivery to TLR9+ human monocyte-derived dendritic cells (DC). Subsequently, DC internalizes and presents the antigen to T cells, inducing their differentiation into effector T cells. KLK alone evokes a sustained adaptive Th2 immune response. Additionally, KLK enhances antigen binding to antigen-presenting cells (APCs), forming a vaccine-specific library at the site of TB vaccination [45]. Although KLK alone elicits a Th2 immune response, its combination with ODN1a results in a more intense Th1 immune response [46,47].

### 3.2. Nucleic Acid Adjuvants

In vaccine research, specific nucleic acids, notably CpG DNA, have demonstrated adjuvant properties as TLR9 agonists. CpG DNA, an oligomer of unmethylated cytosine and guanine deoxyribonucleotides [48], significantly enhances antigen presentation and triggers a specific immune response in the anti-TB subunit vaccine, GamTBVac. Acting as a robust adjuvant, CpG DNA prompts a Th1 immune response, fostering CD4+ T cell and IFC-c production. It boosts both humoral and cellular immune responses by activating TLR9, stimulating antigen-presenting cells and hastening immune responses [49,50]. Moreover, CpG DNA rapidly triggers B cells to secrete IL-6 and IL-10. B cells respond independently, leading to B cell proliferation and their differentiation into antibody-secreting cells. CpG ODN also induces DC activation to produce IL-12, IL-6 and TNF-α. Notably, IL-4 and IL-13 negatively regulate CpG ODN-induced DC cytokine production. In the presence of IL-12, the DC–T cell interaction favors a Th1 response. CpG ODN induces the generation of CD4+CD25+ cells through plasma cell-like DC (pDC) activation, leading to the secretion of IL-10, TGF-β, IFN-γ and IL-6, thereby exerting a potent immunosuppressive effect [51]. Furthermore, CpG ODN enhances MHC, CD40 and CD86 expression on peripheral blood cells, thereby improving antigen processing and presentation [52].

### 3.3. Liposomal Adjuvants

Liposomes, as antigen carriers for subunit vaccines, directly impact the body’s immune response by modifying antigen properties such as charge, composition and size [53]. Amphipathic lipids typically create laminar phases and, when balanced with excess water, form closed vesicles, predominantly composed of aqueous-separated bilayers from various lipids (known as multilamellar liposomes). These liposomal adjuvants serve as delivery systems for antigens and immunostimulants. Liposomal flexibility enables different molecules, such as lipid-based immunostimulants and protein-based antigens, to bind within the same liposomal dispersion [54]. AS01E, containing monophosphoryl lipids (MPL) and saponin QS21, acts as an immunostimulant [55]. Both MPL and QS21 form the core of the GSK adjuvant, offered either as water-in-oil emulsions (AS02) or liposomes (AS01) [56]. Utilizing this liposomal adjuvant, the M72/AE01E vaccine targets tuberculosis. Multiple studies have found that AS01E triggers robust responses, involving CD4+ T cells, bridging Th1 and Th2 cytokines and activating CD8+ T cells and NK cells [57]. MPL, a low-toxicity lipopolysaccharide synthetic variant, serves as a TLR4 agonist to adjuvinate AS01 and AS02. MPL is approved for use in HPV and HBV vaccines [58]. MPL stimulates TLR4 through MyD88 and TRAM/TRIF pathways to enhance inflammatory responses. Moreover, QS-21 induces the production of IL-2, IFN-γ and IgG2a antibodies [59]. M72/AS01E prompts human peripheral blood mononuclear cells (PBMCs) to release substantial M72-specific T cell cytokines, notably high levels of Th1 cytokines (IFN-γ, TNF-α and IL-2) and low levels of IL-17. This activation leads to NK cells producing antigenic IFN-γ and elicits an immune response from CD8+ T cells [57].

Additionally, the cationic liposome adjuvant, CAF01, utilized in human clinical trials, was integral in the development of the anti-tuberculosis vaccine, H1:CAF01 [60]. CAF01 primarily comprises dimethyldioctadecylammonium (DDA), a cationic liposome composed of a hydrophilic dimethylammonium head group attached to two hydrophobic 18-carbon alkyl chains [61]. In aqueous media, DDA spontaneously segregates into dual vesicle layers, facilitating antigen transfer [62]. DDA is positively charged and can easily attach to cell surfaces abundant in negative charges. It also binds to negatively charged proteins and DNA, effectively transporting them into APC cells [63]. Moreover, DDA facilitates fusion or cross-presentation with endosomal membranes, enabling antigen delivery into the cytoplasm [64].

### 3.4. Other Potential Novel Adjuvants

Various adjuvants, beyond the previously outlined categories, are under exploration and application. Chitosan, a hydrophilic molecular carrier, belongs to natural polymer materials with good biocompatibility, degradability, low toxicity and stability [65,66]. It has recently garnered significant attention due to its capacity for good cellular uptake and adhesion, prolonged release duration, continual stimulation of the body’s immune system, effective antigen-presenting cell uptake, adjuvant/immune-boosting impact and inhibition of antigen degradation in vivo [67]. When coupled with chitosan and its derivatives, novel anti-tuberculosis vaccine candidates augment defense against Mtb infection in mice and produce efficient anti-tuberculosis immune reactions in animal models. Moreover, vaccines utilizing chitosan and its derivatives as adjuvants or delivery systems trigger enhanced immune responses for both enteric and non-enteric immunity. Nonetheless, the non-enteric route may be preferable for administering chitosan-based TB vaccines over the enteric route [68].

Chitosan and mannan are promising adjuvants. Currently, the two main mannans extracted from *Mycobacterium* include lipomannan (LM) and lipoarabinomannan (LAM) [69]. LM and LAM consist of polymeric mannose chains with α (1,6)-mannose repeating units [70]. LM exhibits pro-inflammatory properties, whereas LAM is anti-inflammatory. LM in BCG possesses immunomodulatory characteristics, promoting Th1 cell differentiation for intracellular pathogen immune responses [71]. LM also serves as an adjuvant in the presence of LPS, co-stimulating mouse macrophages, showcasing its adjuvant application potential [72].

For mRNA vaccines, yeast microorganisms have garnered significant attention as delivery systems due to their capacity to elicit effective immune responses and capture and shield vaccine antigens specifically [73]. Yeast has no adverse impact on APC viability post-phagocytosis, differing in cytotoxicity from other organisms like bacteria. It serves as a natural adjuvant with granular properties and safeguards vaccine antigens against biodegradation. mRNA vaccines are more temperature-sensitive than traditional vaccines, given RNA’s inherent instability. External factors such as temperature, pH and enzyme activity can induce rapid RNA degradation [74,75]. Without proper stabilizers, mRNA molecules may be swiftly recognized and cleared by the body’s immune system before entering the target cells [76,77]. Thus, enhancing the thermal stability of mRNA vaccines is an urgent concern. Freeze-dried recombinant yeast vaccines, storable at room temperature, maintain vaccine antigenic capacity for up to one year [78]. Yeast cell walls contain substantial amounts of β-1,3-glucan and mannoprotein, which modulate the body’s inflammatory response and activate cytokines and steroids, promoting specific immune responses. β-1,3-glucans, like mannoproteins, induce Th1 cytokine production and, through interactions with other receptors, facilitate neutrophil recruitment and APC’s specific antigen recognition [79]. Consequently, yeast, as an adjuvant for RNA vaccines, holds great promise, offering a broader avenue for vaccine adjuvant development and utilization in the future. Any adjuvant has two sides, that is, to be used within a safe dose range. Aluminum salt adjuvants are widely used, but their stability changes with variations in temperature and time. Protein adjuvants can effectively enhance the ability of antibody responses and regulate the host immune response. The nucleic acid adjuvant CpG OND is widely used, but some people have some adverse reactions when being vaccinated. The liposome adjuvant is non-toxic and can be degraded in the host, but this expensive adjuvant has poor stability and easily undergoes oxidation. It is difficult for a single adjuvant to induce an ideal immune response, and most currently developed adjuvants are combined to have a synergistic advantage.

## 4. Conclusions

To accomplish the WHO’s goal of “Ending Tuberculosis by 2035”, a deeper understanding of Mtb infection’s immune mechanism can provide a theoretical basis for tuberculosis vaccine development. The rapid development of COVID-19 vaccines offers insights and directions for TB vaccine advancement. Beyond global clinical-stage TB vaccines, various novel ones are undergoing research and development. These include inactivated whole-cell non-tuberculous mycobacterial vaccines; DAR-901 as a booster for the BCG vaccine; the subunit vaccine BG/DPC, which consists of the ferritin BfrB, the heat shock protein GrpE; the emulsifying adjuvant for DPC; and the ID91+GLA-SE vaccine based on the rep-RNA platform [80,81,82]. mRNA vaccines, which have received great focus since the COVID-19 pandemic, have shown substantial advancements against SARS-CoV-2 and influenza viruses. However, progress in mRNA vaccines for respiratory bacterial diseases currently lags behind [83,84]. Phage vaccines are another current focus in vaccine research, as they can elicit cellular and humoral immune responses without infecting human cells or integrating into the human genome. Presently, phage vaccines are primarily used for hepatitis B [85], influenza [86] and anti-tumor applications [87], with limited exploration in bacterial infections. Bacterial drug resistance, a key factor in bacterial infections, could be effectively addressed by phage-constructed vaccines [88]. Nonetheless, a phage vaccine against Mtb remains undeveloped [89]. The mRNA vaccine based on single-stranded RNA phages is a new vaccine developed in recent years. Compared with the traditional mRNA vaccine and adenovirus vector vaccine, the single-stranded RNA phage vaccine is safer and more efficient and can effectively stimulate the humoral and cellular immune responses of the body. Phage MS2 and Phage Qβ are stable at room temperature, and they can effectively solve the problem of the cold chain transportation of mRNA vaccines and reduce the cost [90,91]. In the current research and development of new vaccines, in addition to VPM1002, which has the potential ability to replace BCG, MTBVAC is in the clinical stage as a newborn preventive vaccine; its safety, tolerance and immunogenicity are similar to those of BCG. MTBVAC can induce the body to produce higher levels of CD4+ T cells and is expected to replace BCG. Due to the high cost of vaccine development and insufficient investment in late-stage clinical research, there is no effective drug candidate for late-stage product development. The preliminary results for M72/AS01E are consistent with the WHO’s preferred product characteristics, and further studies are ongoing [2].

The development of new TB vaccines progresses alongside novel adjuvant research. Nano-adjuvants, such as the ginsenosides and oil-in-water nanoemulsions in the hepatitis B vaccine, display stronger humoral and cellular immune-inducing effects compared to traditional aluminum salt adjuvants [92]. Moreover, the Mg/Al-LDH adjuvant, synthesized from layered double hydroxide (LDH) and Mg/Al for the pertussis vaccine, elicits a milder inflammatory response at the injection site than that of aluminum hydroxide (AH) while demonstrating a superior safety profile [93]. Nano-adjuvants hold interest due to their small particle size, targeting ability, low toxicity, prolonged immune response and their capacity to elicit organism-specific cellular and humoral immunity [94].

Mtb, an intracellular bacterium, must activate T cells by delivering antigens before triggering an immune response. Antigen-presenting cells possess numerous pattern recognition receptors on their surface. Consequently, the selection of ligands targeting these receptors as adjuvants is currently a hot research topic and a direction for future adjuvant development. Given the complexity of the Mtb genome, which leads to different infection states upon entering the human body, a single tuberculosis vaccine is insufficient to combat tuberculosis across all populations. To develop effective vaccines, it is essential to choose multiple antigens with diverse components for comprehensive immunoprophylaxis and treatment. When Mtb shifts to starvation conditions or dormancy, its expression of antigens like Ag85B secreted in the early stage of infection decreases, whereas antigen expression induced by latency increases. In antigen selection, the latently associated antigen DosR, with regulatory genes *Rv2029c*, *Rv2031c* and *Rv2627c*, is considered to be a strong T cell antigen that can induce strong humoral and Th1 immune responses [95]. *Rv2031c* (HspX) is one of the most immunogenic antigens associated with Mtb latent infection. HspX can induce cellular immunity and humoral immune responses during the latency period of TB infection. The *hspX* promoter can facilitate the rapid entry of recombinant BCG into DCs [96]. They have the potential to be strong vaccine candidates and drug targets. Although few immunomodulators have been employed in TB vaccines, a plethora of novel adjuvants remains untapped. Ideal vaccine adjuvants should be safe, cost-effective, easy to prepare, efficient and devoid of side effects. Achieving these objectives necessitates in-depth and systematic studies on their mechanisms of action, a comprehensive understanding of their immunological effects and an analysis of the structure–activity relationship of immune adjuvants. These efforts can establish a robust theoretical foundation and provide data support for enhancing vaccine immunological efficacy, thereby fostering research and development and facilitating the clinical transformation of new, efficient, safe and economical adjuvants.

## Figures and Tables

**Table 1 vaccines-12-00038-t001:** Novel clinical candidate TB vaccine.

Vaccine Type	Candidate Vaccine	Purpose of Inoculation	Method of Administration	Clinical Stage	Main Ingredients	Vaccination Crowd
Viral Vector Vaccine	AdHu5Ag85A	Prevention	Administered via aerosol	Ⅰ	Enhances immunity against CD4+ and CD8+ T cells	Healthy volunteers between the ages of 18 and 55 years with a history of BCG vaccination
	MVA85A	Treatment	Intradermal injection	Ⅱa	Induces strong and differentiated CD4+ T cells and increases secretion levels of IFN-γ, TNF-α, IL-2, IL-17 and GM-CSF	Adolescents aged 12–17 years and adults aged 18–49 years with a history of BCG vaccination
	TB/FLU-05E	Prevention	Intranasal injection	Ⅰ	Induces long-term cell-mediated immunity	Healthy volunteers between the ages of 18 and 50 years with a history of BCG vaccination
Live Attenuated Vaccine	VPM1002	Prevention	Intradermal injection	Ⅲ	Activates CD4 and CD8 T cells to induce higher Th 1 and Th 17 activities in CD4 T cells	Neonates at day 0–12 and healthy volunteers aged 18–45 years
	MTBVAC	Prevention	Intradermal injection	Ⅲ	Enhances tumor antigen-specific CD4 and CD8 T cell responses	Neonates at day 0–7
mRNA Vaccine	BNT164a1	Prevention	Intradermal injection	Ⅰ		Healthy volunteers between the ages of 18 and 55 years
	BNT164b1	Prevention	Intradermal injection	Ⅰ		Healthy volunteers between the ages of 18 and 55 years
Subunit Vaccine	ID93+GLA-SE	Prevention	Intradermal injection	Ⅱa	Triggers a multifunctional CD4+ T cell response, primarily generating IgG1 and IgG3 subclass antibodies	Healthy volunteers between the ages of 19 and 64 years with a history of BCG vaccination
	AEC/BC02	Prevention	Intradermal injection	Ⅱa	Induces a Th 1 response, increases IFN-γ levels and activates CD8+ T cells	Healthy people over 18 years old and latent infected people
	M72/AS01E *	Treatment	Intradermal injection	Ⅱb	Induces specific CD4 T cell responses	HIV patients and latent infected people aged 16–35 years
	H56:IC31	Prevention	Intradermal injection	Ⅱb	Induces CD4+ T cell responses	Tb patients aged 18–60 years
	GamTBVac	Prevention	Intradermal injection	Ⅲ	Induces specific and persistent Th 1 immune responses and humoral immune responses	High incidence population of tuberculosis aged 18–45 years with a BCG vaccination history

* Represents the most promising vaccine.

**Table 2 vaccines-12-00038-t002:** Adjuvants commonly used in TB vaccines.

Adjuvant	Composition	Adjuvant Type	Signal Channel	Mode of Action	Candidate Vaccine	References
GLA-SE	GLA in a stable oil-in-water SE	Liposomal	TLR4	Induces a Th1 immune response and produces IFN-γ, IL-2 and TNF-α as well as both IgG 1 and IgG 3	ID93+GLA-SE ID91+GLA-SE	[26,27,28]
BC02	CpG DNA, Al(OH)3	Aluminum salt	TLR9	Induces Th1 and Th2 immune responses, producing IL-4 and inducing the expression of CD83, CD86 and MHC-II molecules	AEC/BC02	[29,30]
IC31	KLK, ODN1a	Protein	TLR9	Induces either Th1or Th17 immune responses, producing IFN-γ, IL-2 and TNF-α	H56:IC31 H1:IC31	[42,43,44,45]
CpG ODN	CpG ODN	Nucleic acid	TLR9	Induces a Th1 immune response, producing IFC-c, IL-6, IL-10, IL-12, TGF-β, IFN-γ and TNF-α; promotes the surface expression of MHC, CD40 and CD86 on peripheral blood cells	GamTBVac	[46,47,48,49,50,51,52]
AS01E	MPL, QS-21	Liposomal	TLR4	Induces a Th1 immune response, producing IFN-γ, TNF-α and IL-2 and low levels of IL-17	M72/AS01E	[53,54,55,56,57,58]
CAF01	DDA, TDB	Liposomal	Mincle	Induces a Th17 immune response, producing IL-17 and IgA	H1:CAF01	[59,60,61,62,63]
Chitosan	Chitosan			Activates the inflammasome, releases pro-inflammatory cytokines and ultimately activates Th1, Th2 and Th17 immune responses		[64,65,66,67]
Mannosan	Mannosan			Activates NK-κB, and the inflammasome induces the Th1 immune response and produces TNF-α, IL-12, IL-6 and IL-1β		[68,69,70,71]

## Data Availability

Not applicable.

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
