# Peer review of "Enhancing TB Vaccine Efficacy: Current Progress on Vaccines, Adjuvants and Immunization Strategies"

_vaccines, 2023, doi:10.3390/vaccines12010038_

Round 1

Reviewer 1 Report

Comments and Suggestions for Authors

This is highly educational review of the most current directions for the enhancement of TB vaccine efficacy. Authors thoroughly and systematically list the most important and novel candidates for TB vaccines being tested at different stages of clinical trials, they also discuss strategies in adjuvant applications and immunization strategies.

This review seems to be highly educational for medical practitioners as they summarize some of the exciting observations from the basic in vitro sciences and from the pharmaceutical side of vaccine development. Despite using multiple acronyms for vaccine candidates, the whole reads well and tables are very helpful.

Conclusion does list underdeveloped strategies (mRNA and phage vaccines) which may be next strategy. However, the reader may want to learn which of the vaccines is in the most advanced stages of development; most of them are in the I or II phase of clinical trials and one can wonder which vaccine currently in trials was designed to address the shortcoming of BCG vaccine s which is relative lack of efficacy in adolescent and adults, and they don’t treat infected individuals.

Author Response

Dear Reviewer:

Reviewer 2 Report

Comments and Suggestions for Authors

The article by Hui Wang et al. reviews the current status of new tuberculosis vaccine formulations. The article is divided into two parts: one for the different vaccine approaches (vector, live attenuated, mRNA or subunit) and the other dedicated to adjuvants.
The article is well written and easy to understand.
Minor comments
- It would have been desirable to have a table showing current trials.
- Also, a figure explaining the state of the art.
- I regret that the therapeutic/immunotherapeutic vaccine approach is not the subject of discussion.

Author Response

Comments 1: [It would have been desirable to have a table showing current trials.Also, a figure explaining the state of the art.]

Response 1:  Thanks for your suggestions and we have added the Table 1 to address and summary the novel clinical TB vaccine candidates.

Comments 2: [I regret that the therapeutic/immunotherapeutic vaccine approach is not the subject of discussion.]

Response 2: Thanks for your suggestions. According to the requirements of the column, we mainly discussed the adjuvant of vaccines. In terms of content, we mainly emphasize the types of vaccines, immune pathways and therapeutic effects, and the application of adjuvants.

Reviewer 3 Report

Comments and Suggestions for Authors

The review article ‘Enhancing TB Vaccine Efficacy Current Progress on Vaccines, Adjuvants, and Immunization Strategies’ by Wang et al provides a comprehensive overview of the progress of TB vaccine efficacy and use of adjuvants and immunization strategies. I have summarized my suggestions below to further improve this review.

The Introduction has good context but authors may consider adding a concise statement about the importance of TB vaccine efficacy in the context of TB as a global health burden. Authirs may include relevant citations to support the additional information.

Authors have written a separate section on different types of vaccine which is truly informative to the readers. A detailed note on adjuvants is comprehensive but it would be ideal to add a brief summary by comparing strengths and weaknesses of different adjuvants types mentioned in this review article. Please also add a brief explanation of each vaccine type mentioned and highlight their mechanisms of action and specific advantages of each methodology including adsorption process in the body.

Authors should include information on the most recent developments in TB vaccine research and provide a summary of the key findings till date and based on these findings what are the potential future directions for TB vaccine discovery? Also, describe the main takeaways from the current state of TB vaccine research and what areas further require more focus. Authors have detailed few examples along with adjuvants but consider adding a graphics or table whichever is appropriate to visually represent the information and enhance readability. For example authors can include a table summarizing different vaccines and their stages of limitations and development till date.

Comments on the Quality of English Language

Moderate english check is required

Author Response

Dear Reviewer:
